# Line Drawing Extraction from Cartoons Using a Conditional Generative Adversarial Network

**Kyungho Yu, Juhyeon Noh and Hee-Deok Yang \***

Department of Computer Engineering, Chosun University, 309 Pilmun-Daero, Dong-Gu, Gwangju 61452, Korea; infinitegh@chosun.kr (K.Y.); narak@chosun.kr (J.N.)
\* Correspondence: heedeok_yang@chosun.ac.kr

**Abstract:** Recently, three-dimensional (3D) content used in various fields has attracted attention owing to the development of virtual reality and augmented reality technologies. To produce 3D content, we need to model the objects as vertices. However, high-quality modeling is time-consuming and costly. Drawing-based modeling is a technique that shortens the time required for modeling. It refers to creating a 3D model based on a user's line drawing, which is a 3D feature represented by two-dimensional (2D) lines. The extracted line drawing provides information about a 3D model in the 2D space. It is sometimes necessary to generate a line drawing from a 2D cartoon image to represent the 3D information of a 2D cartoon image. The extraction of consistent line drawings from 2D cartoons is difficult because the styles and techniques differ depending on the designer who produces the 2D cartoons. Therefore, it is necessary to extract line drawings that show the geometric characteristics well in 2D cartoon shapes of various styles. This paper proposes a method for automatically extracting line drawings. The 2D cartoon shading image and line drawings are learned using a conditional generative adversarial network model, which outputs the line drawings of the cartoon artwork. The experimental results show that the proposed method can obtain line drawings representing the 3D geometric characteristics with a 2D line when a 2D cartoon painting is used as the input.

**Keywords:** line drawing; conditional generative adversarial networks

## 1. Introduction

Content created using 3D (three-dimensional) computer graphics technology is used in various fields, such as games, movies, animation and education, because it allows users to experience a real-world environment with 3D content. Unlike 2D (two-dimensional) content, 3D content allows the user to interact directly with content components that feel immersive and realistic. Recently, with the development of virtual reality and augmented reality technologies, it has become easier for people to access 3D content. This ease of access increases people's interest in 3D content.

To produce 3D content, we need to make 3D models. These models express elements constituting the content as vertices in the 3D space. However, high-quality 3D modeling is time-consuming and costly. The drawing-based modeling technique is commonly used to reduce the time required to perform 3D modeling. This technique involves creating a 3D model based on line drawings or sketches [1–4].

There are two ways to make cartoon images. The first one is to draw the detailed parts themselves (e.g., materials and shades) to produce a cartoon image. The second one is to use computer graphics technology in which a shape is created by 3D modeling. The texture of the object is displayed through texture mapping. The cartoon images are then created through cartoon rendering by using toon shading [2–4].

Feature line extraction from a 2D cartoon image, such as Canny edge detection in a 2D cartoon image, is different from the characteristic lines generated according to the curvature of the 3D model or the relationship between each vertex and the viewpoint. As

can be seen in Figure 1a, the feature lines extracted from the 2D cartoon image do not represent the geometric properties of the 3D model. On the other hand, the feature lines extracted from 3D model represent the properties of the 3D model, as shown in Figure 1b. The purpose of this paper is to extract 3D feature lines from a 2D cartoon image, as shown in Figure 1c.

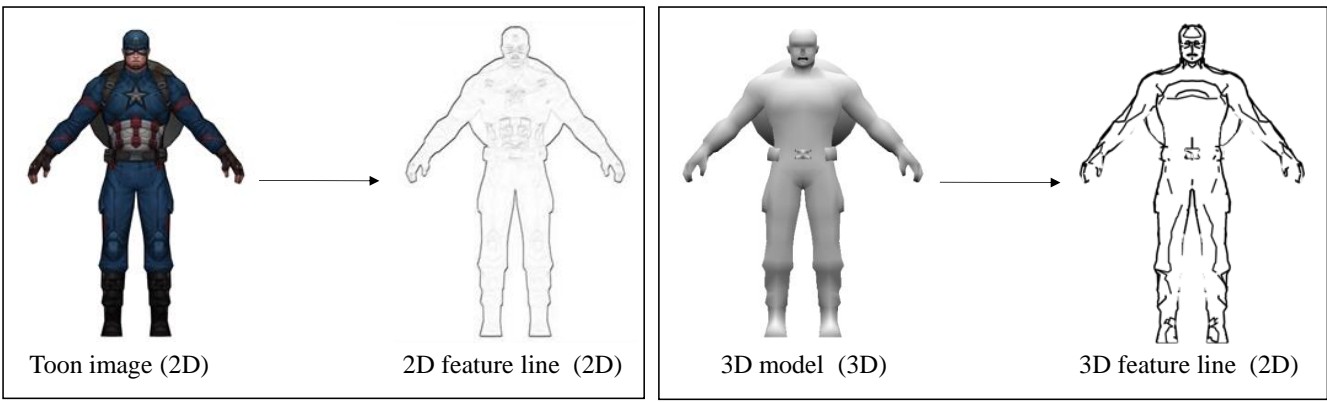

(a) A 2D feature extraction from a 2D toon image  (b) A 3D feature extraction from a 3D model

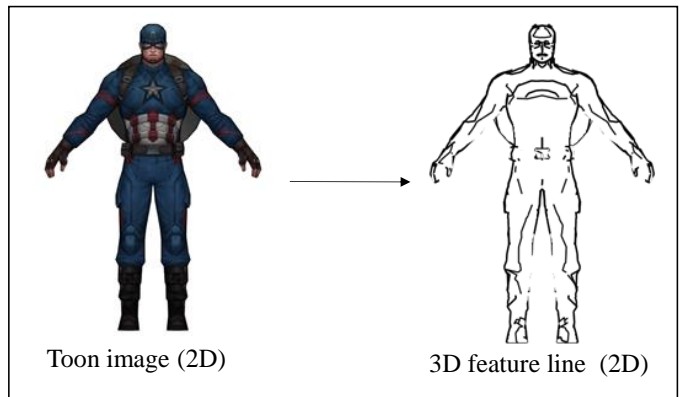

(c) A 3D feature extraction from a 2D toon image

**Figure 1.** Feature lines extracted from the 2D images and 3D models.

This paper proposes a method for automatically extracting line drawings from cartoons using a conditional generative adversarial network (GAN) model, as shown in Figure 2. The conditional GAN model can be trained using 2D cartoon shading images and line drawings. The line drawings are automatically extracted when a 2D cartoon object of various styles is inputted using the trained model. Figure 3 shows the process of extracting line drawings from the original 2D cartoon painting.

In the first step, a toon shading image and a line drawing image are created in the 3D model as a dataset production step. We use toon shading to create a 3D model that looks similar to a cartoon to create a cartoon image. To create a line drawing image, we generate it using various feature lines representing the geometric properties of the 3D model. The line drawings are combinations of the feature lines. In the second step, the dataset is learned using the conditional GAN model. In the third step, the learning result is evaluated using a test set.

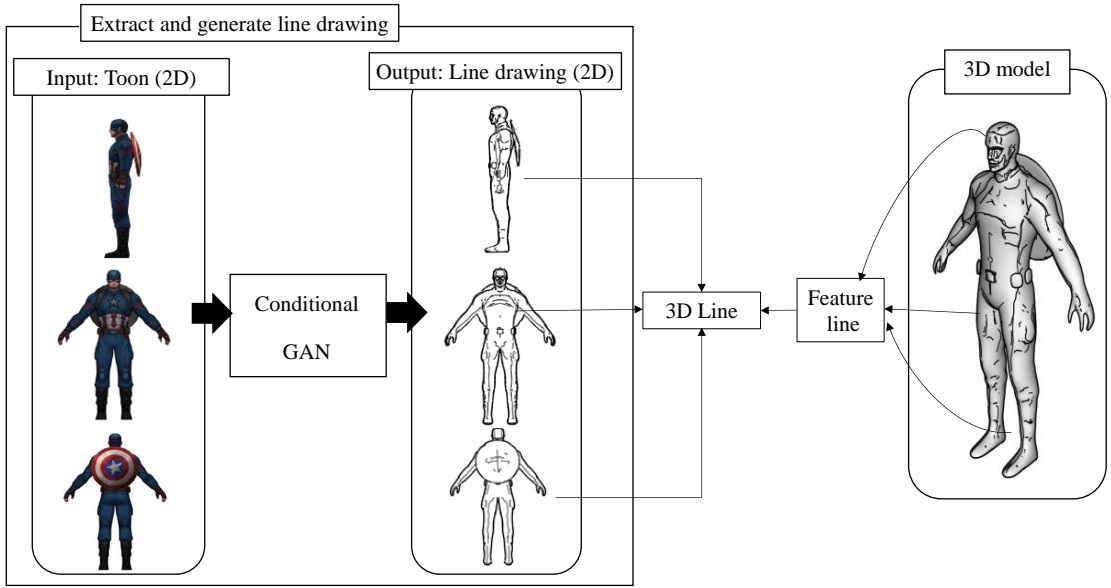

**Figure 2.** Feature detection in the visible geometry.

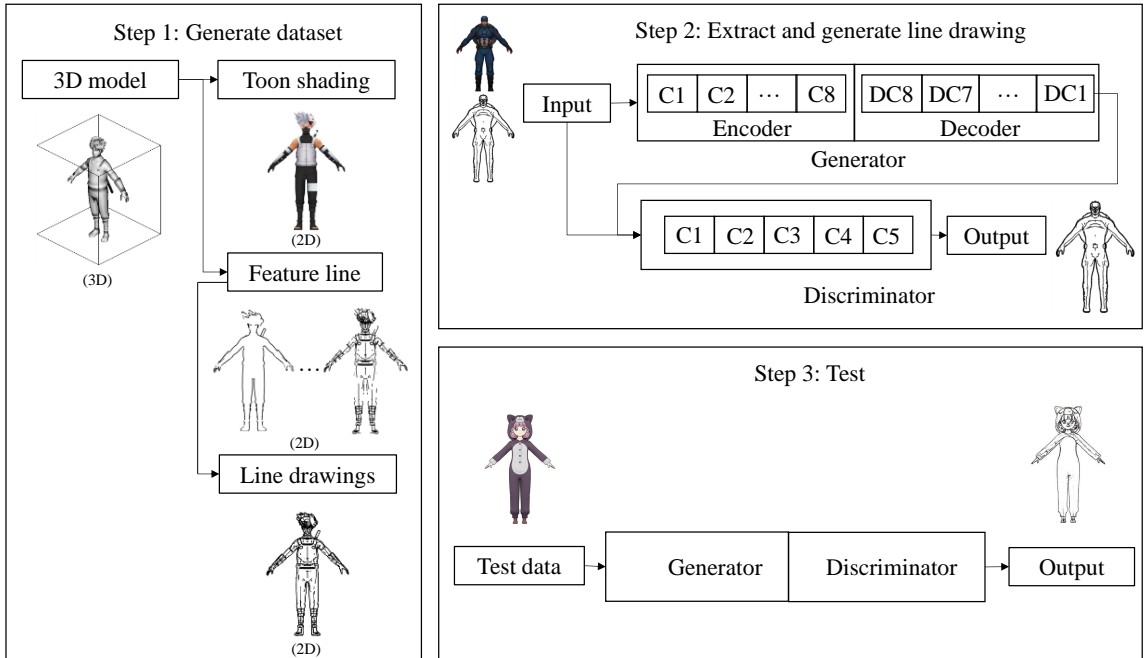

**Figure 3.** Structure of the proposed method.

## 2. Related Works

### 2.1. Geometric Feature Lines

2.1.1. Types of Feature Lines

A feature line determines where to draw a line in an object. The types of feature lines are contours, suggestive contours, ridges, valleys and apparent ridges. As the concept of these feature lines has been established, studies have been conducted on the efficiency of feature extraction and how the feature lines may be drawn [5–8]. As shown in Figure 4, feature lines differ according to the geometric properties and the shape of the 3D model. Therefore, depending on human perception, these feature lines are evaluated for a good representation in esthetic and realistic depictions. The results show that suggestive contours and apparent ridges are generally good at portraying features [9].

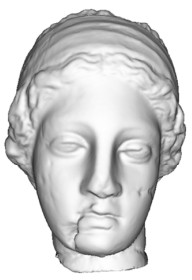 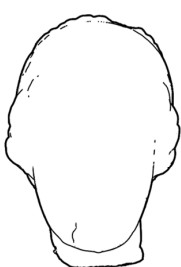 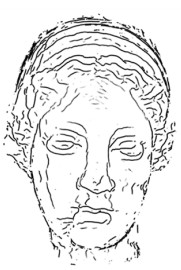 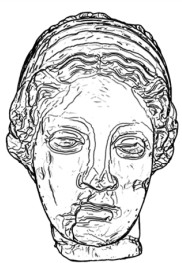 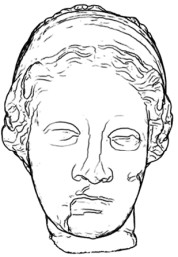

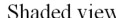
Shaded view       Contours       Suggestive contours       Ridges & Valleys       Apparent ridges

**Figure 4.** Comparison of the various feature lines [7].

### 2.1.2. Contours and Suggestive Contours

Contours are view-dependent lines that represent the points created by the contour generator and are projected onto a 2D image.

The contour generator is a set of points satisfying Equation (1), where point $p$ is on surface $S$. $n(p)$ is the normal vector of point $p$ and $v(p)$ is the view vector from point $p$ to viewpoint $c$ [5–8].

$$n(p) \cdot v(p) = 0. \tag{1}$$

Contours cannot represent the details in a shape. There are suggestive contours as feature lines that can be used with contours to represent the details in a shape and effectively provide the shape information. Suggestive contours can represent feature lines in detail that are not shown by contours [5,6].

Suggestive contours are lines drawn on clearly visible parts of the surface, where a true contour would first appear with a minimal change in viewpoint [5,6].

### 2.1.3. Ridges, Valleys and Apparent Ridges

Ridges and valleys are view-independent feature lines. They are sets of points whose principal curvature in the main direction is locally maximum. At point $m$, the curvature operator $S$ is defined by Equation (2):

$$S(r) = D_r n \tag{2}$$

where $n$ is a normal vector and $D_r n$ is a directional differential coefficient in the vector r direction in the tangent plane.

For all points on the surface, the maximum and minimum principal curvatures $k_1$ and $k_2$ are the eigenvalues of $S$ when $|k_1| > |k_2|$. Ridges and valleys are sets of points with the direction derivative coefficients $D_e k_1 = 0$ in vectors $e^1$ and $e^2$ that match the directions of the maximum and minimum principal curvatures. Ridges have $k_1 > 0$ whereas valleys have $k_2 < 0$.

Apparent ridges are view-dependent lines and a set of points whose view-dependent curvature is the largest in the view-dependent principal direction. Given an object $M$, such that $M \subset R^3$ where the viewing screen plane is $V$, and a point $m$, such that $m \in M$, then point $m'$ can be obtained by projecting point $m$ onto the viewing screen plane $V$. Given a point $m'$, such that $m' \in V$, the view-dependent curvature operator $Q$ at point $m'$ is defined by Equation (3) [7]:

$$Q(s) = D_s n' \tag{3}$$

Here, $D_s n'$ is a directional differential coefficient that differentiates $n'$ in the vector $s$ direction in the screen plane. $n'$ is the normal vector of point $m$ projected onto the viewing screen plane. $Q(s)$ is a vector of the tangent planes that indicates how the normal vectors change as they move along vector $s$ on the screen.

### 2.2. Autoencoder

An autoencoder trains the output value to make it as close as possible to the input value $x$. The structure of the autoencoder consists of three layers: an input layer, a hidden

layer and an output layer. The number of hidden layers is smaller than that of the input layers. Moving the data from the input layer to the hidden layer is called encoding whereas moving them from the hidden layer to the output layer is called decoding.

The input vector $x$ is encoded as a hidden representation $y$ using Equation (4) where $s$ is the activation function, W is the $d' \times d$ dimensional weight matrix and $b$ is the bias vector. The hidden layer representation $y$ is decoded into an output vector $z$ using Equation (5).

$$y = f_\theta(x) = s(Wx + b). \tag{4}$$

$$z = g_{\theta'}(x) = s(W'x + b'). \tag{5}$$

The autoencoder is optimized using a loss function to reduce the difference between the input and the output. The mean squared error (MSE) is used when $z$ is a continuous value and the cross-entropy (CE) function is used when a binary classification is used. The smaller the value of the loss function, the smaller the error and the closer the output value to the input value.

$$L_{MSE}(x,z) = \|x - z\|^2 \tag{6}$$

where $z$ represents the output value. The MSE can be defined as the square of the difference between the input and the output value using Equation (8).

The CE function is defined by Equation (7):

$$L_{CE}(x,z) = -\sum_{k=1}^{d} [x_k(log(z_k)) + (1 - x_k)log(1 - z_k)] \tag{7}$$

where $k$ is the $k$th vector data and $d$ is the dimension.

### 2.3. 3D Shape Extraction

Drawing-based modeling can be roughly classified into two types. The first is to create a 3D model based on the drawing. The second is to compare the drawing of the 3D model from the input drawing and the nearest 3D model retrieved from a database and modify it to fit the input drawing [1,2].

Lun et al. proposed an architecture to infer a 3D shape that is consistent with sketches from one or more views of an object. Their method is based on a convolutional neural network (CNN) trained to map sketches to 3D shapes. Their method can generalize to reconstruct 3D shapes from human line drawings that can be approximate, noisy and not perfectly consistent across different viewing angles [3].

Recent work has employed CNNs for predicting surface depth and normal from real images [9,10]. Driven by the success of encoder–decoder architectures [11–14] that can effectively map inputs from one domain to another, newer methods use such architectures with convolutions in three dimensions to generate 3D shapes in a voxelized representation [15–17].

### 3. Line Drawing Extraction Using a Conditional GAN

#### 3.1. Generation of Line Drawings with a 3D Model

In this section, we show how a line drawing image is created by combining the various feature lines described in Section 2.1. The reason for this is that the portion representing the shape of the model is different for each characteristic line and that the understanding of the shape through the line is different for each person. The combination of feature lines consists of contours, suggestive contours, ridges, valleys, apparent ridges and boundaries.

Figure 5 shows an example of extracting feature lines from a 3D character model. Figure 5b is a line drawing using contours, ridges, valleys and boundaries whereas Figure 5c is a line drawing using contours, suggestive contours, ridges, valleys, apparent ridges and boundaries. Figure 5c shows better geometric characteristics in the arms, coat and abdomen than those shown in Figure 5b. In Figure 5b,c, no line is drawn inside the arms whereas the legs are marked with dotted circles because the vertices of this part are not detected as feature lines.

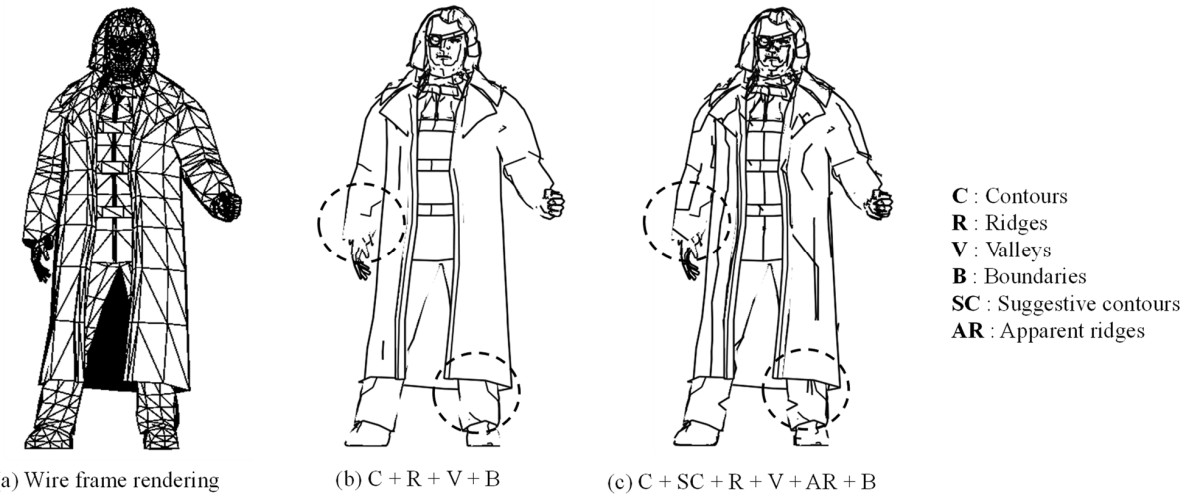

| | |
|---|---|
| (a) Wire frame rendering | (b) C + R + V + B |
| | (c) C + SC + R + V + AR + B |

**C** : Contours
**R** : Ridges
**V** : Valleys
**B** : Boundaries
**SC** : Suggestive contours
**AR** : Apparent ridges

**Figure 5.** Line drawing as a combination of the feature line.

To solve this problem, we performed a subdivision surface on the 3D model before the feature line extraction. Figure 6 shows that the number of vertices in the 3D character model increases as a result of the subdivision surface and that the surface is smoothed [18].

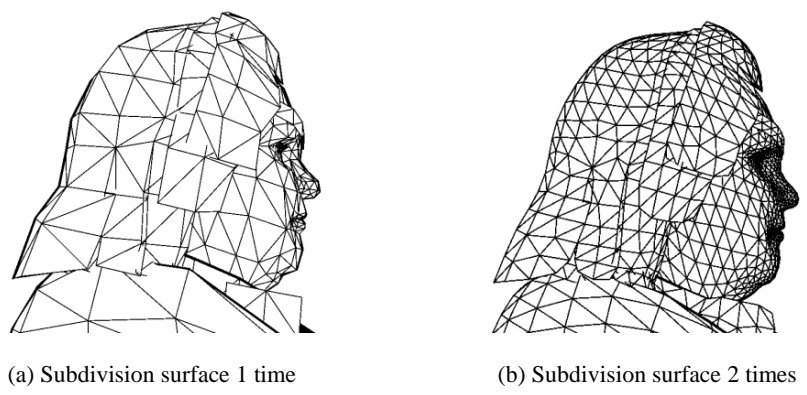

(a) Subdivision surface 1 time          (b) Subdivision surface 2 times

**Figure 6.** Subdivision surface of the 3D model.

Figure 7 shows the result of the line combination according to the number of subdivision surfaces. When the number of subdivision surfaces is compared, it can be seen that there is no line in the shape part (dotted line rectangle) of the coat in Figure 7b although it can be seen that a line is drawn in accordance with the change in the curvature of the coat (dotted line rectangle). A line is also drawn on the face or the shoe to help recognize the shape. When the combination of the characteristic lines is compared, it can be seen that there is no line in the arm part (solid line rectangle) of Figure 7b although it can be seen that a line is drawn in the arm part (solid line rectangle) of Figure 7c to indicate that the arm is bent. However, in the face part, suggestive contours and apparent ridges are added, making it difficult to recognize the shape. This is because there are many feature lines that express the eyes, nose and mouth with many bends when a 3D model of the face is made. Thus, as shown in Figure 7f, the more the subdivision surface is performed, the more difficult it becomes to recognize the shape. Therefore, to generate the dataset to be used for the learning, we performed the subdivision surface twice in the 3D model to express as many lines as possible. The feature lines use a combination of contours, suggestive contours, ridges, valleys, apparent ridges and boundaries. A combination of feature lines creates a line drawing image in three directions: side, front and back; we used it as a dataset.

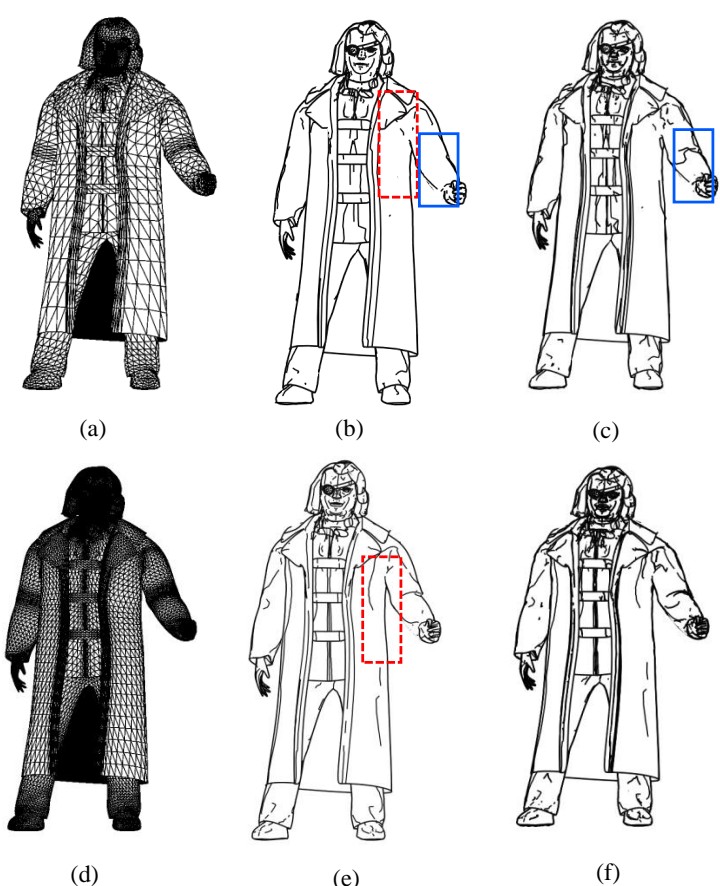

**Figure 7.** Feature line extraction of the 3D model according to the number of subdivision surfaces performed: (**a**) 1 time, wire frame; (**b**) 1 time, first line combination; (**c**) 1 time, second line combination; (**d**) 2 times, wire frame; (**e**) 2 times, first line combination and (**f**) 2 times.

### 3.2. Line Drawing Extraction from a 2D Cartoon Image

In recent years, research has been conducted on the use of deep-learning techniques for drawing-based modeling. In the method of creating a 3D model, the deep-learning technique generates a depth map and a normal map that represent 3D information when a 2D line drawing is inputted.

#### 3.2.1. Generative Adversarial Network

The GAN generates an output similar to that of an autoencoder. The GAN consists of two parts: a generator and a discriminator network. The generator learns by reflecting the distribution of the input data and generates fake data from the random vector whereas the discriminator determines whether the data output from the generator are real or fake data. The objective function of the GAN is given by Equation (8) [19–22].

$$\min_{G}\max_{D}V(D,G) = E_{x\sim p_{data}(x)}[log(D(x))] + E_{z\sim p_x(z)}[log(1 - D(G(z)))] \tag{8}$$

where $G$ is the generator, $D$ is the discriminator, $x$ is the input vector and $z$ is the random vector. The generator tries to determine whether the value of Equation (8) is large so that it can distinguish real data from fake data or small so that it can generate fake data similar to the real data.

#### 3.2.2. Conditional Generative Adversarial Network

A conditional GAN was proposed to generate data by adding a conditional term to the GAN. A conditional GAN can generate data that meet certain conditions. The objective

function of the conditional GAN can be expressed by adding to the objective function of the GAN, which represents a specific condition, as shown in Equation (9) [19–22].

$$\min_G \max_D V(D,G) = E_{x \sim p_{data}(x)}[log(D(x|y))] + E_{z \sim p_x(z)}[log(1 - D(G(z|y)))].\quad(9)$$

First, the conditional GAN learns by using a generator network consisting of convolution and deconvolution layers. The generator consists of eight convolution layers and eight deconvolution layers, each one using a U-net network, as shown in Figure 8. In the encoding process (convolution), key features are detected while reducing the size of the feature map to understand the image. According to the feature map after the end of the encoding, the cartoon data are finally restored as line drawing data through decoding (deconvolution). Skip connections are used to restore the lost information in this process.

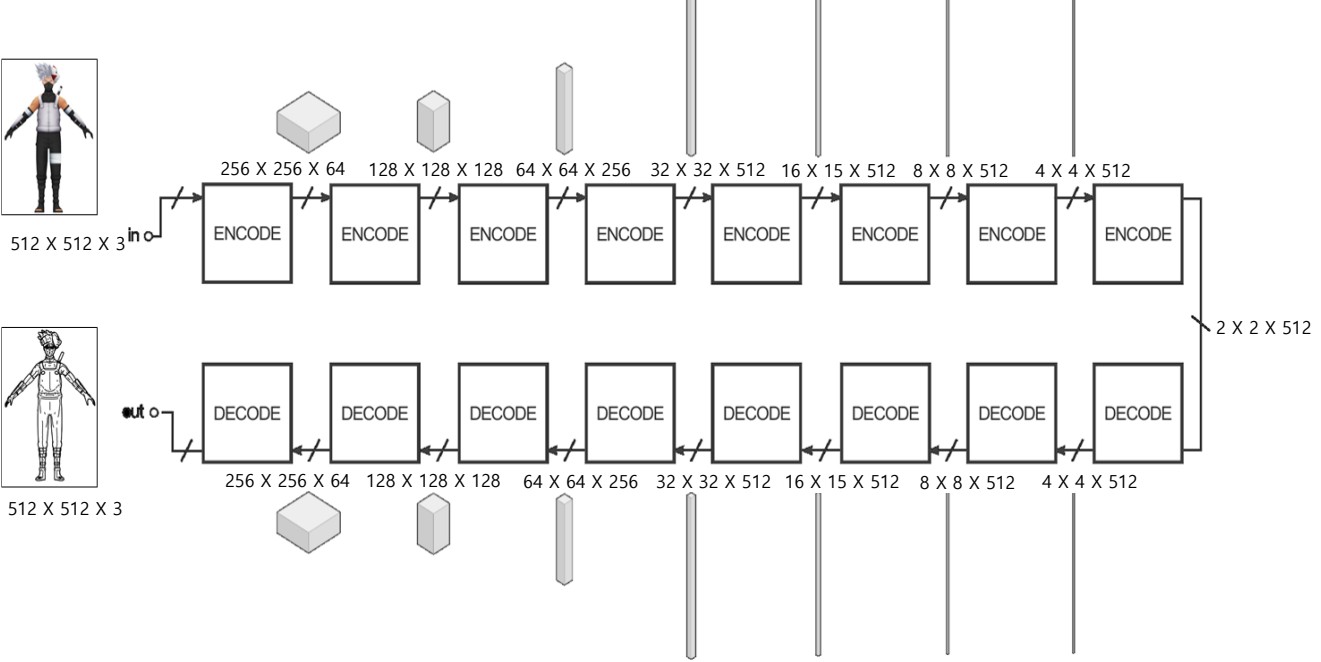

**Figure 8.** The structure of the generator.

The discriminator creates a patch on the basis of the input image and determines the input using this patch. The discriminator uses the patch GAN to determine whether the image generated by the generator corresponds with the input image. It does not determine whether the whole image is genuine or fake but determines whether a patch size is of a specific size and then it averages the result [19–22]. The structure of the discriminator is shown in Figure 9.

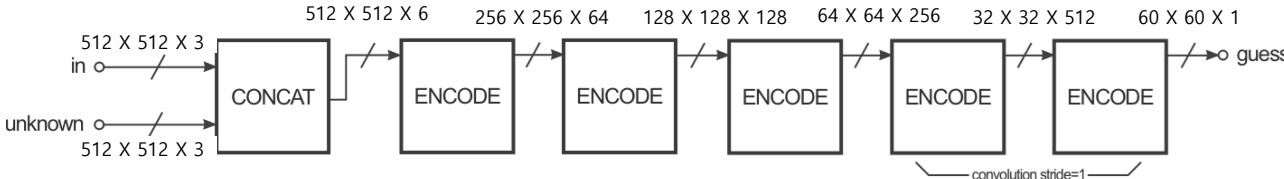

**Figure 9.** The structure of the discriminator.

The input data consist of a 2D cartoon image $x$ and a corresponding line drawing $y$. The generator $G$ learns the inputted 2D cartoon image, as shown in Figure 10, and generates fake data $G(x)$ similar to the line drawing. The discriminator $D$ alternately learns the 2D cartoon image $x$, the real data $y$ and the fake data $G(x)$ generated from the generator

and it determines whether $y$ and $G(x)$ are genuine data or not. Equation (10) is the objective function of the conditional GAN used.

$$\min_{G}\max_{D}E_y[log(D(y))] + E_x[log(D(1 - D(G(x)))] + E_{x,y}[\|y - G(x)\|].$$ (10)

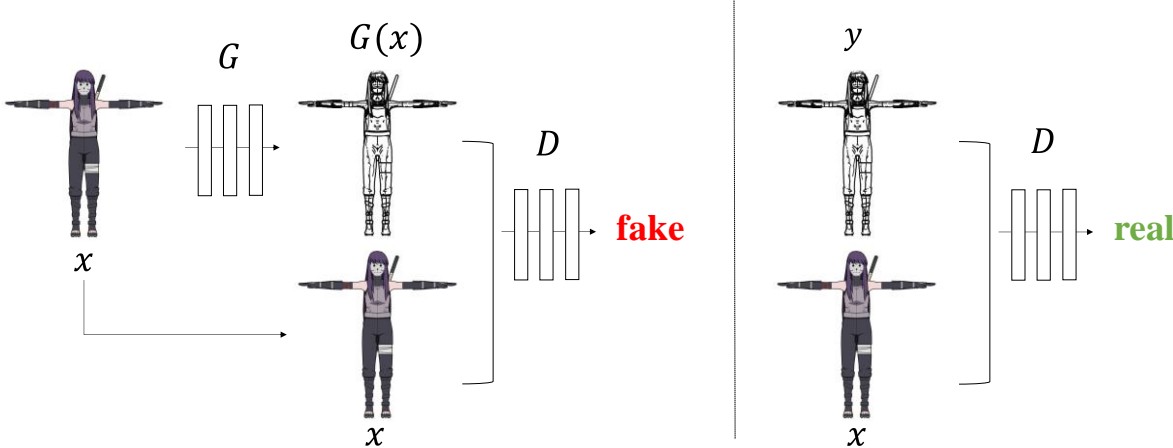

**Figure 10.** Conditional GAN training for line drawing extraction from 2D cartoon images.

As shown in Equation (10), the generator needs to generate an image that is as close as possible to the inputted 2D cartoon image so that the result of the objective function is minimized. The discriminator has the maximum value of the objective function to determine whether the line drawing generated by the generator corresponds with the 2D cartoon image. The learning is repeated by optimization based on the value of the objective function.

The discriminator can distinguish whether the line drawing generated by the constructor at the beginning of the learning is the data corresponding with the 2D cartoon image. However, as the learning progresses, the generator generates fake data $G(x)$ near $y$, making it difficult to distinguish whether it is genuine or fake. By inputting a 2D cartoon image through this iterative learning, we can generate data that are close to the actual line drawing.

Feature line detection and generation are performed by modifying pix2pix [12] using a conditional GAN. pix2pix is a network that generates images from images and the existing pix2pix generates color images from edge images. However, we modified pix2pix to detect line drawings in cartoon images for this study.

## 4. Experimental Results and Analysis

### 4.1. Experimental Environments

Table 1 shows the experimental environment and design for evaluating the proposed method. The datasets used for the learning consisted of cartoon images (Figure 11a) and their corresponding line drawings (Figure 11b) created from 896 three-dimensional models. The image size was $512 \times 512$ pixels. The line drawing used in the study consisted of a subdivision surface twice larger than that of the 3D model. The feature lines consisted of a combination of contours, suggestive contours, ridges, valleys, apparent ridges and boundaries. We also performed experiments with a dataset without a subdivision surface to determine the effect of the subdivision surface. We used 10-fold cross-validation for the evaluation. The test was performed by increasing the training time by 100 units from the 200th to the 400th.

Cartoon images and their 3D models were collected based on a database published on the internet and the collected data were modified for the purpose of the experiment [23].

**Table 1.** Experimental environment and design.

| Deep-Learning Model | Conditional GAN |
| --- | --- |
| Experimental environment | CPU: i7-8700K; GPU: 1080 Ti; RAM: 11 GB |
| Dataset | Frontal view: 896 images |
| Number of training times | 200 to 400 times (increased by 100 units) |
| Feature line | Contours, suggestive contours, ridges, valleys, apparent ridges and boundaries |

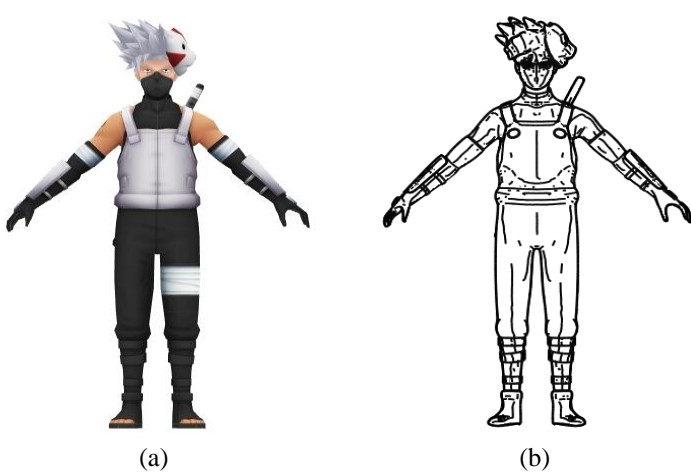

(a)  (b)

**Figure 11.** Example of training data: (**a**) cartoon image and (**b**) line drawing.

### 4.2. Experimental Results and Analysis

A visual analysis of the experimental results of this study showed that the geometric attributes of the 3D model, which were not seen in the input data, appeared in the output line drawing when the 2D cartoon image was inputted. The arm, thigh and abdomen parts of the 2D cartoon image (shown in Table 2) were difficult to visually recognize on the surface curvature. However, the curvature of the output line drawing could be visually recognized by expressing the curvature of the surface as a line.

The performance result of the subdivision surface of the 3D model was visually analyzed. Table 2 shows the results of learning the dataset and outputting it without performing the subdivision surface and the dataset wherein the subdivision surface was performed twice. The result of performing the subdivision surface twice showed that the geometric properties looked better visually in the chest, abdomen and leg regions than the geometric properties obtained when no subdivision surface was performed. The reason for this result was that the subdivision surface increased the curvature of the 3D model and, consequently, many feature lines were represented.

Tables 3–5 show the results of the line drawing extracted from a 2D cartoon image. As shown in Tables 2–5, as the number of subdivisions increased, the number of vertices in the 3D character also increased and the surface of the 3D character was smoothed. As the surface of the 3D character was smoothed, the line drawing features (such as contours, suggestive contours, ridges, valleys, apparent ridges and boundaries) were more extracted. As more features were added in the feature lines, the feature lines became more complicated. This made a few regions that had many vertices in 3D model unusable. For example, the outputs of the face regions were too complex to recognize the components of the face, as shown in Tables 2–5.

**Table 2.** Output according to the number of subdivision surfaces performed.

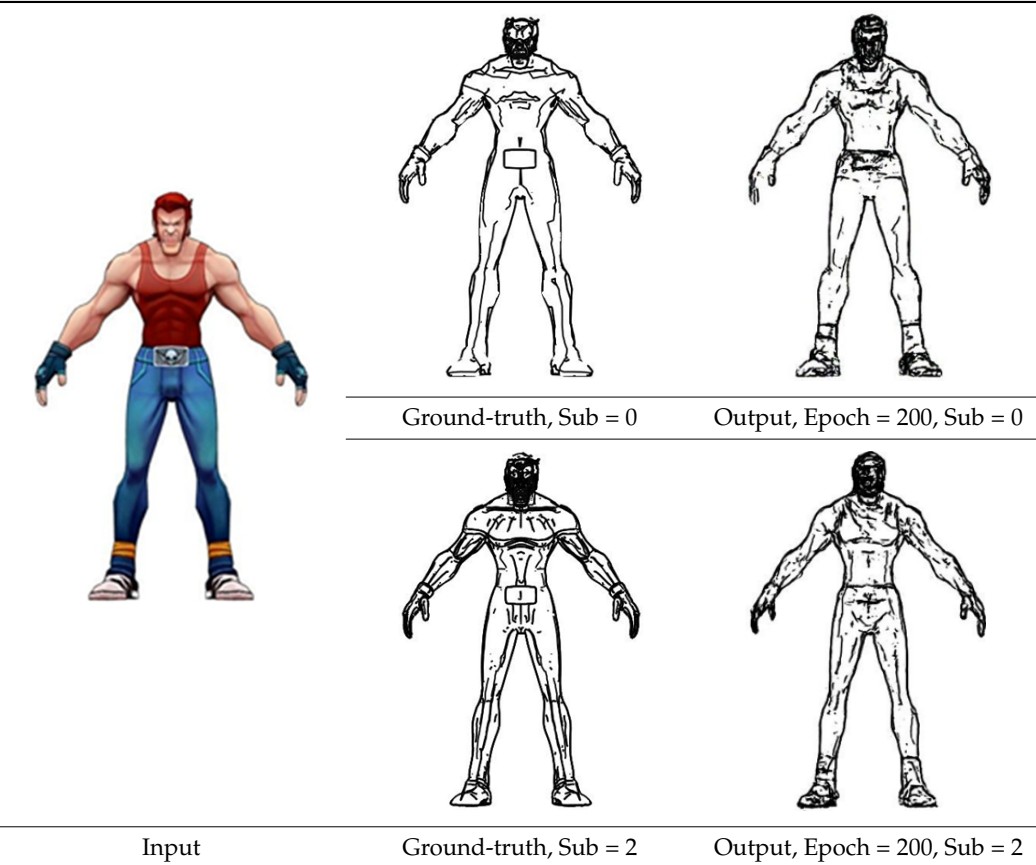

| Input | Ground-truth, Sub = 2 | Output, Epoch = 200, Sub = 2 |
|---|---|---|

**Table 3.** Result of the line drawing extraction using a conditional GAN (Example 1).

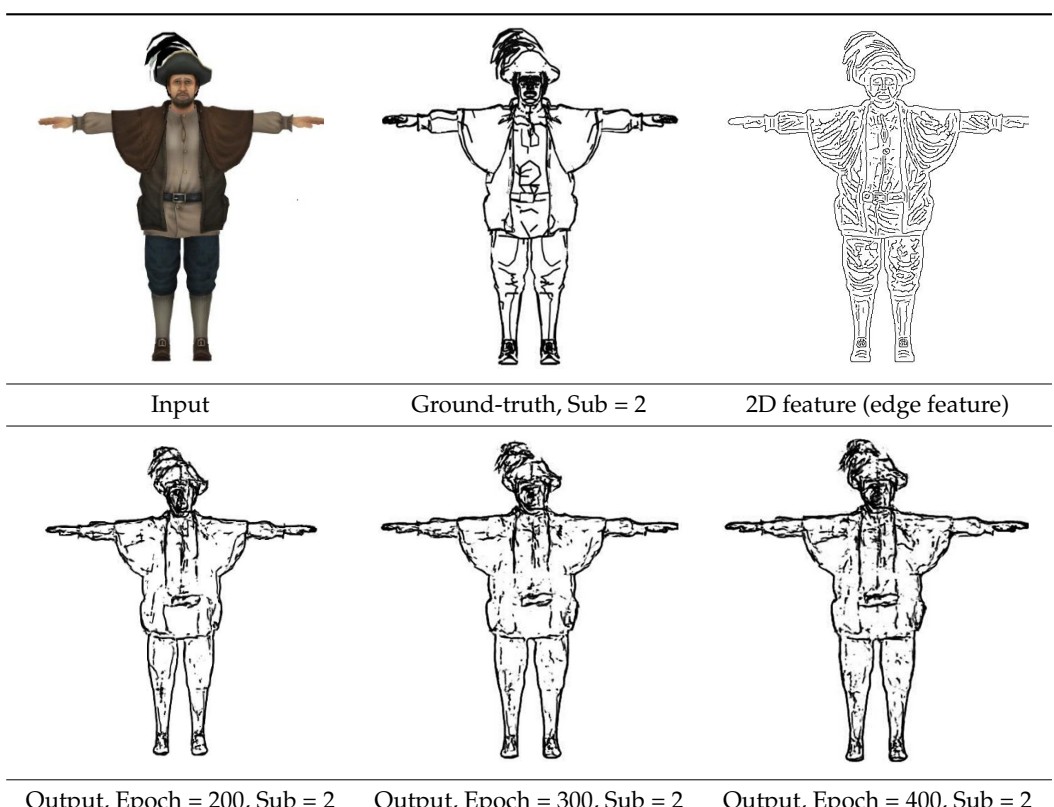

| Output, Epoch = 200, Sub = 2 | Output, Epoch = 300, Sub = 2 | Output, Epoch = 400, Sub = 2 |
|---|---|---|

**Table 4.** Result of the line drawing extraction using a conditional GAN (Example 2).

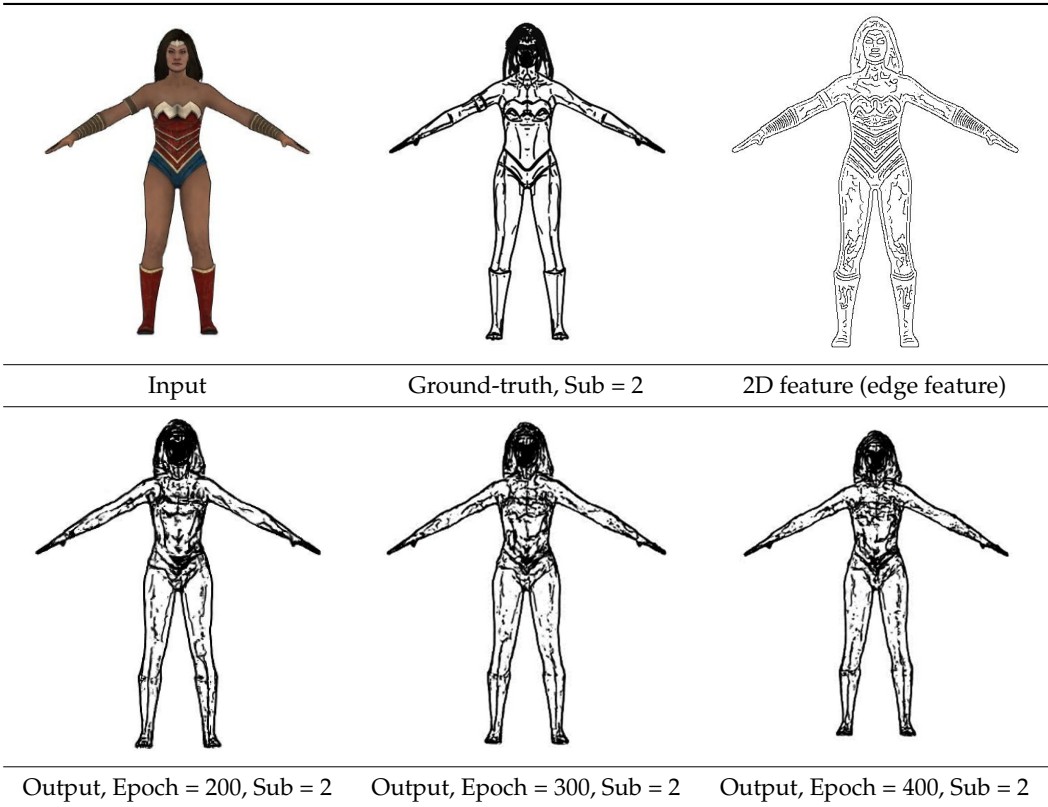

| Input | Ground-truth, Sub = 2 | 2D feature (edge feature) |
| --- | --- | --- |
| Output, Epoch = 200, Sub = 2 | Output, Epoch = 300, Sub = 2 | Output, Epoch = 400, Sub = 2 |

**Table 5.** Result of the line drawing extraction using a conditional GAN (Example 3).

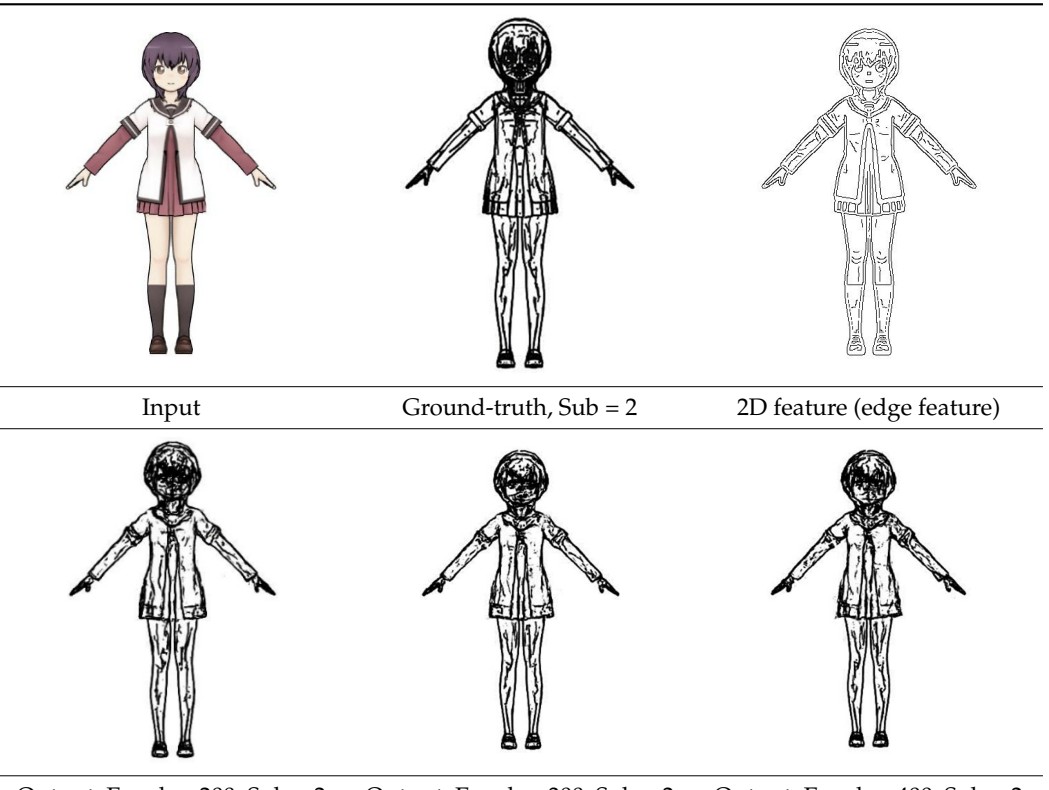

| Input | Ground-truth, Sub = 2 | 2D feature (edge feature) |
| --- | --- | --- |
| Output, Epoch = 200, Sub = 2 | Output, Epoch = 300, Sub = 2 | Output, Epoch = 400, Sub = 2 |

The experimental results of this study could not be compared with the results of conventional 2D image generation because the line drawings generated by the proposed method showed the geometric characteristics of a 3D model in a 2D cartoon image using a conditional GAN model. Therefore, the peak signal-to-noise ratio (PSNR) was used for the quantitative measurement. The PSNR is the maximum signal-to-noise ratio, which is numerically expressed in terms of how close the output is to the ground truth. Figure 12 shows the line drawing extraction results according to the epoch number and subdivision surface. As shown in Figure 12, the PSNR decreased as the number of epochs increased and the subdivision surfaces increased.

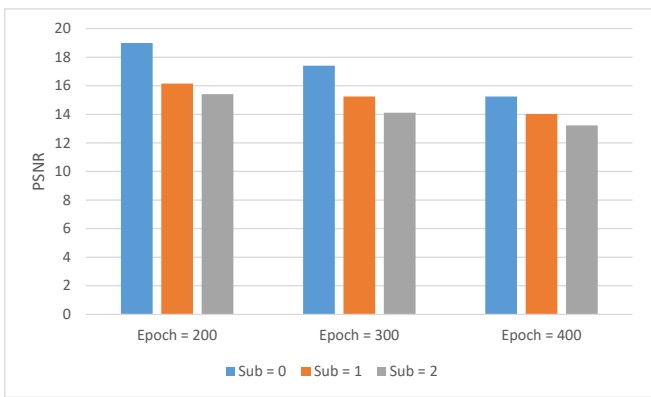

**Figure 12.** PSNRs according to the epoch number and subdivision surfaces.

As shown in Table 6, it can be seen that neither the button nor the water drop shapes on the clothes were created in the output results by confusing the button and water drop shapes on the clothes. The trained model recognized a button as a shape and did not generate information about it.

**Table 6.** Result of the line drawing extraction using a conditional GAN (Example 4).

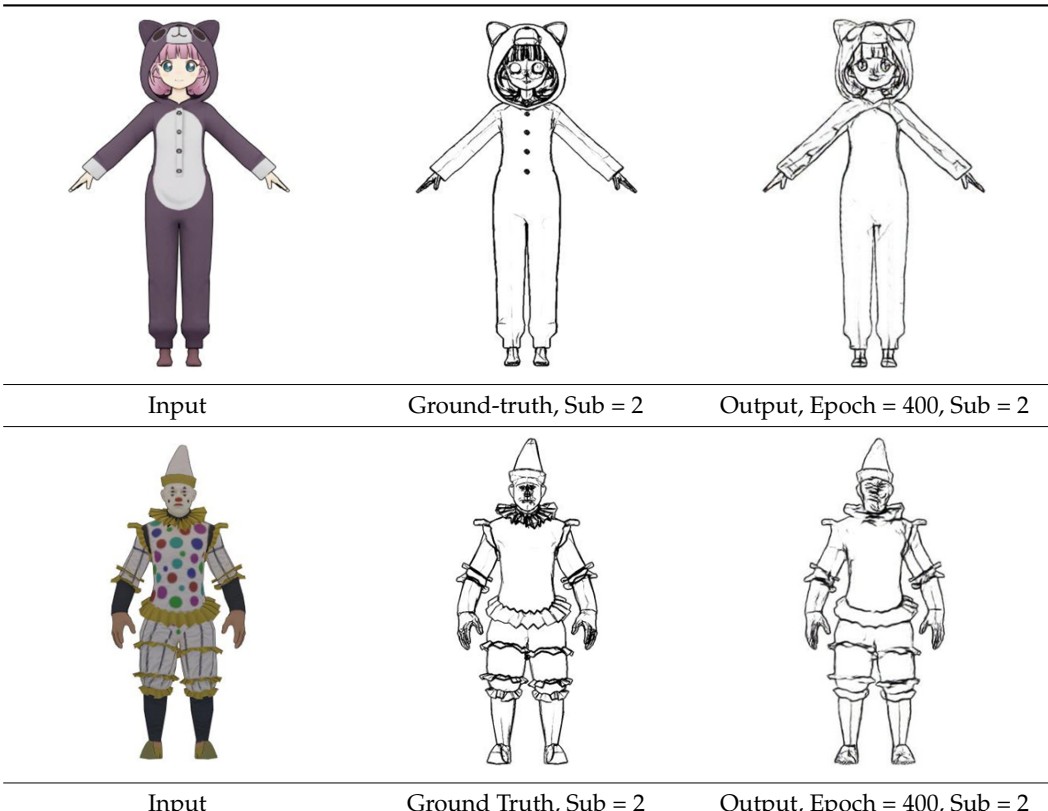

Table 7 shows the output results of extracting line drawings using only the face region. As shown in the results, the face components were well extracted compared with the face regions of Tables 2–5.

**Table 7.** Result of the line drawing extraction using a conditional GAN using the face region.

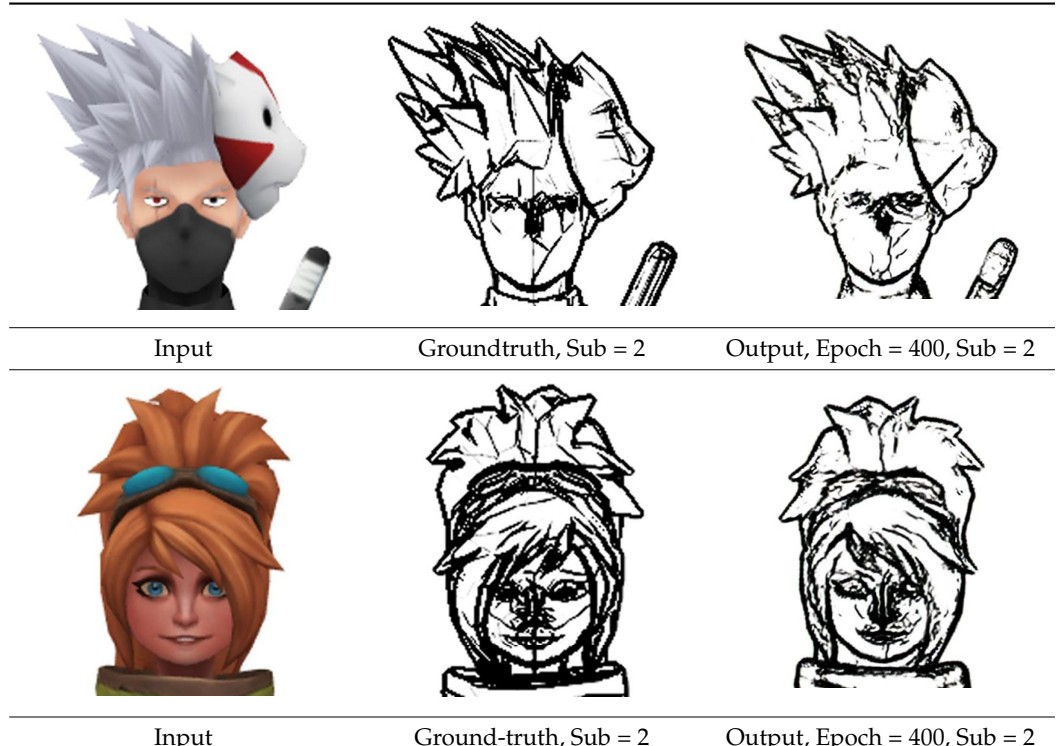

| Input | Groundtruth, Sub = 2 | Output, Epoch = 400, Sub = 2 |
| --- | --- | --- |
| Input | Ground-truth, Sub = 2 | Output, Epoch = 400, Sub = 2 |

## 5. Conclusions and Further Work

In this paper, we proposed a method for extracting line drawings automatically from 2D cartoons using a conditional GAN model. The proposed method extracted a line drawing by combining a 3D model with a subdivision surface twice to generate a dataset and by combining contours, suggestive contours, ridges, valleys, apparent ridges and boundaries. The generated dataset was then learned by the conditional GAN model using a 2D cartoon image and a line drawing. Finally, a test was performed by increasing the number of learning times from 200 to 400 in increments of 100 and then the results were confirmed by using a validation set after the learning.

The results of the experiment confirmed that the line representing the shape was drawn more accurately when the subdivision surface was performed than when it was not performed. When the number of learning times was increased, the number of lines was increased and the drawing was clearly drawn. As a result, it was confirmed that the geometric attributes that were not seen in the 2D cartoons were represented by the line drawing of the experimental results of this study.

A limitation of this study is the consistency problem of the 3D model used to generate the dataset. The 3D model used in this study was inconsistent in the extraction of the line drawing because of the different numbers of polygons representing the accuracy. In other words, in the 3D model made with few polygons, the feature line was not detected but was detected by performing a subdivision surface. On the other hand, the 3D model made with many polygons suffered from the over-detection of the feature line by the subdivision surface. To solve this problem, we extracted line drawings on the basis of the number of subdivision surfaces performed.

In future work, we will conduct experiments using actual cartoon paintings and we will create a 3D model from 2D cartoon images using the line drawings generated by

the proposed method for use in drawing-based modeling. If the results of the proposed method are used as inputs to a 3D model generator, it is expected that a 3D model in which the shape of the character is expressed in detail can be produced compared with the models in existing studies using contours and suggestive contours as the inputs.

**Author Contributions:** K.Y. worked in the conceptualization, simulation, validation and manuscript writing of this paper. J.N. worked in the validation and contributed to the final version of the manuscript. H.-D.Y. worked in the manuscript writing of this paper and supervised the funding acquisition. All authors have read and agreed to the published version of the manuscript.

**Funding:** This work was supported by the National Research Foundation of Korea (NRF) grant funded by the Korea government (MSIT) (No. NRF-2017R1A2B4005305, No. NRF-2019R1A4A1029769).

**Institutional Review Board Statement:** Not applicable.

**Informed Consent Statement:** Not applicable.

**Data Availability Statement:** Not applicable.

**Conflicts of Interest:** The authors declare no conflict of interest.

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
