# Peer review of "Line Drawing Extraction from Cartoons Using a Conditional Generative Adversarial Network"

_applsci, doi:10.3390/app11167536_

Round 1

Reviewer 1 Report

Dear Authors,

Thank you for submitting your manuscript on the subject of line drawing extraction from cartoons (or here: toon-rendered imagery) using conditional GANs. I think the topic or objective of generating 3D geometry from 2D artistic (instead of photogrammetric) imagery is very incentive and has lots of potential of application, among which you as authors already mention already AR/VR. Also the specifically addressed approach of deriving contour feature lines (for 3D reconstruction) from 2D toon renderings via conditional GANs is interesting and timely, using top-notch knowledge of the deep-learning community. The topic is also located in computer graphics for entertainment, and the great coloured illustrations certainly fit the tone of the manuscript and the subject.

That said, there are some more or less severe drawbacks in this manuscript. I hereby outline the major points, while detail comments on content and especially language are left to the commented paper. First of all, the introduction starts with a template-like structural statement of how to write a paper introduction. In good faith, I assume this statement is a left-over from internal manuscript revision – still, it does not belong there at all. Secondly, the paper overall exhibits a bad structure, which I will outline separately below. Thirdly, there is too little information about the actual deep learning and the neural network, in terms of its layout, the specific feature-size progression, its learning metrics such as learning- and decay rate, and so forth. Fourthly, the paper lacks a dedicated discussion section, which is reinforced by the lack of an actual post-experiment discussion. Points I think specifically require critical discussion and commenting are the impact of render image size, the impact of localized mesh refinement / subdivision, the impact of the data size (i.e. number of unique training data), the method’s applicability to human-generated cartoon input, and the application to actual 3D model reconstruction. Fifth and last, the paper employs a questionable error metric (PSNR, and its formulation) and shows a dubious convergence behaviour (derived from PSNR plot).

The term of “bad structure” in this case refers to the following points:

a) the already mentioned template statement in the introduction (first paragraph)

b) the introduction itself lacks some essential (motivational) information of why CGI is used for training instead of actual artistic concepts (e.g ‘paintings’)

c) instead of the above-mentioned motivation, the introduction contains a confusing section about what non-photorealistic rendering (NPR) is and what it is used for, which is out-of-context at the point it is mentioned. As NPR is also not the area in graphics this paper makes its contribution to, this part is out-of-place in the introduction.

d) the related work is more of a textbook explanation and textbook overview of concepts rather than a topic-specific, pinpoint literature review of the used concepts and papers and their shortcomings to the research being presented. This means that large section of the related work are out-of-context.

e) the second half of the method section is again an extended textbook overview. This means its out-of-context and - as part of the literature - in the wrong place of the paper.

f) instead of the textbook review passage in the method section (e), the authors need to explain in more detail how their neural network is layed out and designed, for example with a commonplace neural network architecture plot with annotations. The authors further need to explain the neural network meta-parameterization and the learning setup, such as optimisation algorithm, learning target, convergence target and convergence rate.

g) from the method section, it is not apparent what the actual contribution of the paper is:

g.1) is it the feature line drawing generation ? If so, then - as a concept - it is not novel.

g.2) is it the neural network itself ? possibly novel, but the authors do not provide enough actual information about their network to assess that.

g.3) is it about the whole pipline and the system as-a-whole ? Then not only is the neural network part lacking depth and details, but then the method section also needs to explain the software design, how that is novel, how it integrates those building blocks of image generation and learning into one, and how the system could also be adapted to interact with the user - after all, the purpose of the method (as given by the authors), is 3D digital content creation (DCC) for entertainment purposes (e.g. games, movies, etc.) that use AR/VR, so the interactive part seems to be important.

h) the experimental results insufficiently states the key metrics about the dataset(s) being used (e.g. number of images used for train/test/validation; comparison to other networks or approaches; where does the dataset come from ?). The authors use very pretty example illustrations, but the result section lacks analytical depth.

All of those points are structural deficiencies where either a part of the text is out-of-context, or it is out-of-place, or both, or a section lacks structurally the depth it is meant to cover.

Again, the here outlined points are the main revision requirements, but the revision is not only limited to those but also to all the other moderate and small issues covered in the commented manuscript.

Kind Regards,

The Reviewer

Author Response

Please, refer to the attached file.

Reviewer 2 Report

This paper presents a line extraction method from cartoons using neural networks. The work is very interesting, and the applicability is undeniable. However, I have several comments regarding the presentation of the research:

  1. There are several oversights in the paper:
    • The authors forgot to exclude the instructions from the template in the Introduction section (lines 27 to 35)
    • There are a lot of parameters from equations that are either written in italic font, or in normal font – they should always be italic
    • Equations 4, 10, 11, 12: the min and max operators are either written in italic or normal font, starting with lowercase or uppercase letters
    • The same type of line rectangle is named either “dotted” (line 240) or “dashed” (line 242)
    • I am not sure at what Tab. 1 the authors refer to in line 346. I believe they should refer to Table 2 or to Figure 14.
  2. In my opinion, the state of the art in line drawing & 3D shape extraction from 2D images, using neural networks, should be discussed in more detail (maybe in a sub-section of Related Work). This part is merely mentioned in the introduction, with several references that are only inserted, without explanations.
  3. The authors should explain why they chose conditional GANs for the line drawing task, and not other networks.
  4. The subdivision step seems to improve the overall result of the line drawing task, but the lines obtained at head level make the features of the face unusable. This can be observed both in Fig 11f and in all images of Tables 2 – 5 (even in the ground truth images). The authors merely mention this problem in line 249. However, in my opinion, they should address the problem somehow.
  5. I do not understand the explanations of the authors from lines 367-372. What does “lack of consistency between the training set and the validation set” mean? How are two elements from the training and validation sets considered similar?
  6. I do not understand the purpose of Figure 15. It does not convey important information. It just shows the PSNR for the tested images. In my opinion, it would be more interesting to show the PSNR for the tested images, in different conditions, and to see how these conditions affect the result: without the subdivision step, with 1 subdivision, with 2 subdivisions, with 200/300/400 epochs.

Author Response

Please, refer to the attached file.
